# Insights into the cotranscriptional and translational control mechanisms of the *Escherichia coli tbpA* thiamin pyrophosphate riboswitch
Jonathan P. Grondin[1,5], Mélanie Geffroy[1,2,6], Maxime Simoneau-Roy[1,2,7], Adrien Chauvier [1,8], Pierre Turcotte[1,9], Patrick St-Pierre[1], Audrey Dubé[1,10], Julie Moreau[1], Eric Massé [2], J. Carlos Penedo [3,4] & Daniel A. Lafontaine [1] ✉

Riboswitches regulate gene expression by modulating their structure upon metabolite binding. These RNA orchestrate several layers of regulation to achieve genetic control. Although *Escherichia coli* riboswitches modulate translation initiation, several cases have been reported where riboswitches also modulate mRNA levels. Here, we characterize the regulation mechanisms of the thiamin pyrophosphate (TPP) *tbpA* riboswitch in *E. coli*. Our results indicate that the *tbpA* riboswitch modulates both levels of translation and transcription and that TPP sensing is achieved more efficiently cotranscriptionally than post-transcriptionally. The preference for cotranscriptional binding is also observed when monitoring the TPP-dependent inhibition of translation initiation. Using single-molecule approaches, we observe that the aptamer domain freely fluctuates between two main structures involved in TPP recognition. Our results suggest that translation initiation is controlled through the ligand-dependent stabilization of the riboswitch structure. This study demonstrates that riboswitch cotranscriptional sensing is the primary determinant in controlling translation and mRNA levels.

Riboswitches are highly structured regulatory elements located in mRNA untranslated regions[1–8]. These non-coding RNA structures regulate gene expression by undergoing structural changes upon the binding of cellular metabolites[9]. There are more than 55 distinct classes of natural riboswitches that have been discovered to date and even more are predicted to exist[3], suggesting that riboswitches are widespread regulatory elements. In general, riboswitches exhibit one highly conserved element, the aptamer domain, that is directly involved in metabolite sensing[2,10,11]. Due to the large variety of detected metabolites[3], aptamer domains show diverse structures and exhibit high affinity toward sensed ligands[9]. In addition to the aptamer,

riboswitches contain the expression platform domain that is modulating the expression of downstream gene(s). The structure of the expression platform is not well conserved and may regulate various processes such as transcription termination, translation initiation, mRNA decay and splicing[11].

While riboswitches characterized in *Bacillus subtilis* primarily control premature transcription termination[11,12], there is a greater diversity of regulation mechanisms observed for riboswitches in *Escherichia coli*. For example, in silico structure prediction and in vivo reporter gene assays indicate that the *E. coli* thiamin pyrophosphate (TPP)-sensing *thiM* riboswitch strictly modulates translation initiation[13]. However, transcriptional

¹Department of Biology, Faculty of Science, Université de Sherbrooke, Sherbrooke, QC, Canada. ²Department of Biochemistry and Functional Genomics, Université de Sherbrooke, Sherbrooke, QC, Canada. ³Centre of Biophotonics, Laboratory for Biophysics and Biomolecular Dynamics, SUPA School of Physics and Astronomy, University of St. Andrews, St Andrews, UK. ⁴Centre of Biophotonics, Laboratory for Biophysics and Biomolecular Dynamics, Biomedical Sciences Research Complex, School of Biology, University of St. Andrews, St. Andrews, UK. ⁵Present address: Canadian Food Inspection Agency, Ottawa, ON, Canada. ⁶Present address: Delpharm Boucherville, Boucherville, QC, Canada. ⁷Present address: Cégep de Saint-Hyacinthe, Saint-Hyacinthe, QC, Canada. ⁸Present address: Single Molecule Analysis Group, Department of Chemistry, University of Michigan, Ann Arbor, MI, 48109, USA. ⁹Present address: Unité de recherche clinique et épidémiologique, CIUSSS de l'Estrie, Sherbrooke, QC, Canada. ¹⁰Present address: Département de médecine de famille et de médecine d'urgence, Université de Sherbrooke, Sherbrooke, QC, Canada. ✉e-mail: daniel.lafontaine@usherbrooke.ca

**Fig. 1 | The regulation mechanism of the *E. coli* *tbpA* riboswitch. a** The *tbpA* riboswitch controls the expression of the *tbpA-thiP-thiQ* operon that is involved in the transport of thiamin and related metabolites. The riboswitch is located in the 5' untranslated region and adopts the ON state in the absence of TPP in which the anti-P1 stem is formed, thereby allowing the initiation of translation. However, in the presence of TPP, the OFF state is adopted in which the P1 stem of the aptamer domain is stabilized, therefore repressing translation initiation. The GUG start codon is shown in blue. **b** Secondary structure of the *tbpA* riboswitch in the TPP-bound OFF structure. The anti-P1 stem is highlighted in yellow and the P1 stem is circled in blue. The Shine-Dalgarno and GUG start codon are shown in blue. **c** Schematics depicting the constructs used for ß-galactosidase assays. All fusions contain the natural promoter. The "- fusion" represents the construct in which the riboswitch domain is not present. The translational fusion (TrL) contains 11 codons of *tbpA* and is directly fused to the *lacZ* reporter. The transcriptional fusion (TrX) contains an additional Shine-Dalgarno (SD) sequence that allows the translation of the *lacZ* reporter without being affected by the structure of the riboswitch. **d** ß-galactosidase assays for the promoter-only (-), translational (TrL) and transcriptional (TrX) fusions. The average values of three independent experiments with standard deviations are shown. ß-galactosidase assays for constructs expressed from an arabinose-inducible promoter (pBAD). Gene expression was assessed for the wild-type (WT), G35C, G121C and U130A constructs in the context of both the translational (TrL) (**e**) and transcriptional (TrX) (**f**) fusions. The average values of three independent experiments with standard deviations are shown.

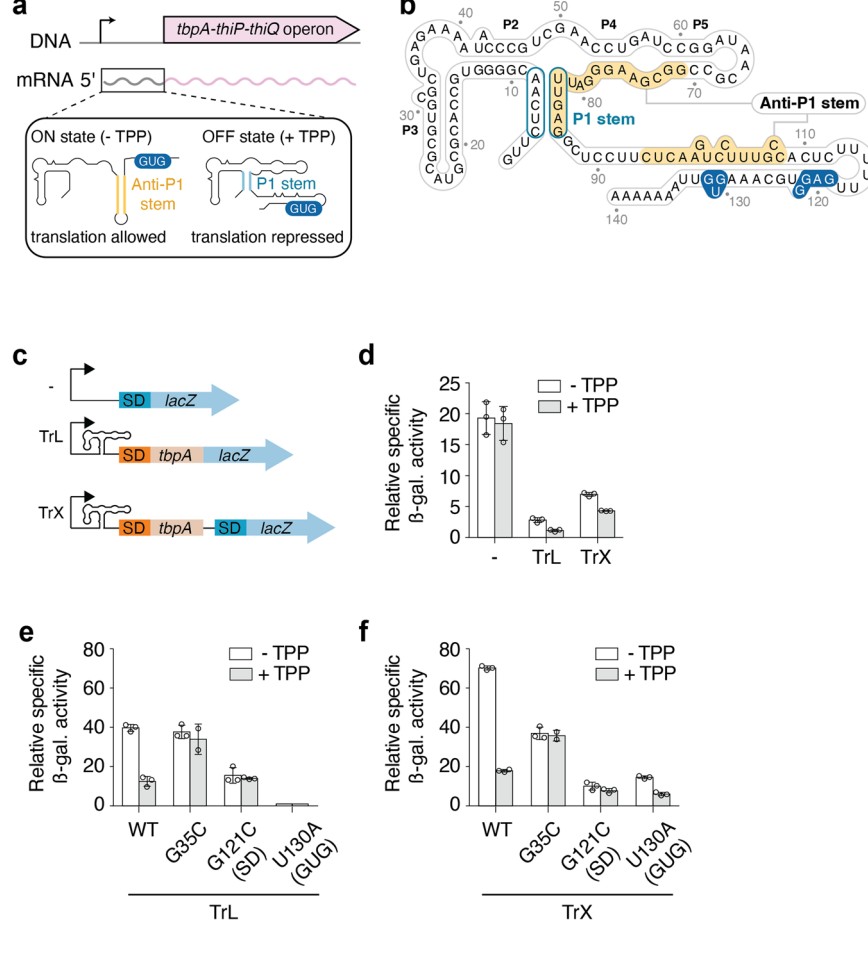

fusions containing at least 34 *thiM* codons revealed a TPP-dependent repression of mRNA levels, consistent with Rho-dependent transcription termination modulated by the efficiency of translation[13]. Similarly, it was found for the TPP-sensing *thiC* riboswitch that both translation initiation and Rho-dependent transcription termination are modulated upon metabolite binding[14]. Importantly, in contrast to *thiM*, it was determined that the Rho binding site is located within the *thiC* riboswitch domain, thereby suggesting that the *thiC* riboswitch encodes the regulatory sequences to control both translation and transcription processes[14]. Lastly, the *E. coli tbpA* TPP-sensing riboswitch, *aka thiB*, is also primarily involved in controlling translation initiation[15–17]. Although it is less studied than the *thiC* and *thiM* riboswitches, the *tbpA* riboswitch was also proposed to regulate Rho transcription termination[13]. A recent study reported that the primary regulatory control used by the *tbpA* riboswitch is at the translational level[17], suggesting a similar mechanism than used by the *thiM* riboswitch.

Several experimental evidence suggest a central role for the cotranscriptional process during riboswitch regulation. For instance, it was demonstrated that the transcriptionally-regulating *B. subtilis* adenine riboswitch poorly achieves ligand binding in the context of the full-length transcript[18,19]. However, efficient adenine sensing was obtained when performed cotranscriptionally, i.e., while the native RNA polymerase (RNAP) is synthesizing the riboswitch sequence[20]. Similar conclusions were obtained for the *B. subtilis* flavin mononucleotide (FMN)-sensing riboswitch, where cotranscriptional FMN sensing relies on several elements such as transcriptional pause sites and the NusA transcription factor[21]. Interestingly, the preference for cotranscriptional metabolite sensing is preserved in translationally-controlling riboswitches, such as the *E. coli* TPP-sensing *thiC*

and *tbpA* riboswitches[14,16,17]. In both cases, the presence of pause sites and the formation of nascent riboswitch structures contribute to ensure efficient cotranscriptional TPP sensing. While the regulatory mechanism of the *thiC* riboswitch has been characterized, less is currently known regarding how cotranscriptional sensing of the *tbpA* riboswitch is harnessed to modulate the initiation of translation.

Here, using a combination of in vivo and in vitro assays, we show that the *tbpA* riboswitch exerts a control at the translational level. Our data further indicate that the *tbpA* mRNA levels are also decreased in the presence of TPP, consistent with Rho-dependent transcription termination[13]. In agreement with *tbpA* controlling translation initiation, toeprint assays show that the binding of the 30S ribosomal subunit is prevented upon TPP recognition. Our data revealed that TPP sensing is more efficient when occurring cotranscriptionally than post-transcriptionally and that the speed of transcription is important for ligand sensing, in agreement with previous data[16]. Lastly, smFRET experiments show that structural dynamics is observed at the level of the aptamer domain, consistent with both the aptamer and expression platform being structurally coordinated in the response to TPP.

## Results
### The *tbpA* riboswitch regulates both translation and mRNA levels
The *tbpA* riboswitch controls the expression of the *thiBPQ* operon that encodes an ABC transporter system involved in thiamin and TPP transport (Fig. 1a)[15]. The *tbpA* riboswitch is predicted to regulate at the level of translation initiation, which is accomplished by selectively sequestering the Shine-Dalgarno (SD) sequence upon TPP binding (Fig. 1b). In the absence

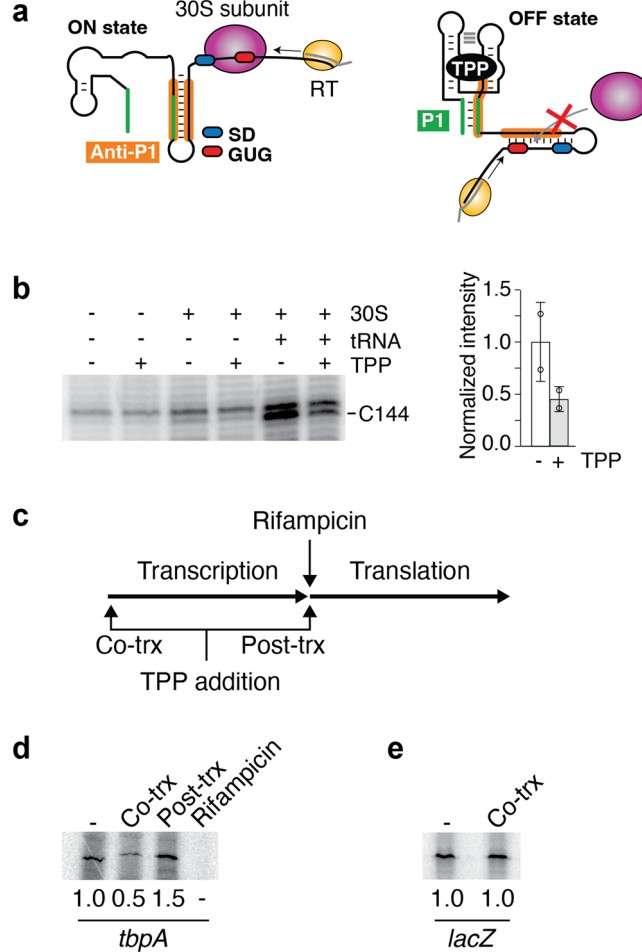

**Fig. 2 | The *tbpA* riboswitch modulates translation initiation through cotranscriptional TPP sensing. a** Schematics representing the toeprint assays. *Left*, in the absence of TPP, the formation of the ON state allows the binding of the 30S ribosomal subunit. In such a case, the reverse transcriptase (RT) is expected to stop elongating upon reaching the bound 30S subunit. The Shine-Dalgarno (SD) and the GUG start codon are shown in blue and red, respectively. The anti-P1 stem is shown in orange. The DNA oligo used to produce the cDNA is shown in gray. *Right*, in the presence of TPP, the formation of the OFF state prevents the binding of the 30S subunit, thereby not producing a toeprint at the ribosome binding site. The P1 stem is shown in green. **b** *Left*, toeprint assays performed on the *tbpA* transcript as a function of the 30S subunit, tRNA-fMet and TPP. The toeprint at position C144 is preferentially obtained with the 30S subunit and tRNA-fMet, but is decreased upon TPP binding. *Right*, quantification of the toeprint efficiency. The experiments have been performed three times and the average and the standard deviations are shown. **c** Experimental assays monitoring TPP binding using in vitro transcription-translation assays. In these assays, transcription is first allowed to proceed until the addition of rifampicin. The second step involves the addition of amino acids to permit translation. The effect of TPP on the translation efficiency is assessed by adding it during the first (cotranscriptional) or second (post-transcriptional) step. **d** Transcription-translation assays assessing TPP binding to the riboswitch. Reactions were achieved in the absence (-) or presence of TPP added either cotranscriptionally (Co-trx) or post-transcriptionally (Post-trx) as indicated in **c**. The ratio of repression is shown below the gels. **e** Control experiments for transcription-translation assays were performed with *lacZ*. No translation repression was obtained when adding TPP cotranscriptionally. The ratio of repression is indicated below the gels.

of TPP, the structure of the riboswitch folds the anti-P1 stem, which disrupts the aptamer domain and exposes the SD sequence to promote translation initiation (Fig. 1a).

We first investigated the *tbpA* regulatory mechanism using *lacZ* reporter gene assays. To determine whether the expression is regulated at the promoter level, we engineered a fusion in which *lacZ* was directly fused at the endogenous promoter (Fig. 1c). When growing cells in a minimum media without and with 1 mM TPP, no regulatory effect was observed on the activity of the promoter (Fig. 1d), suggesting that the latter is not regulated by TPP levels. We next engineered a translational construct in which *lacZ* was fused after the 12th codon of *tbpA* (Fig. 1c). When using this construct, we observed that the expression of *tbpA* was reduced by ~6.3-fold compared to the promoter-only construct (Fig. 1d). In the presence of TPP, it was observed that *lacZ* expression was decreased by ~2.3-fold (Fig. 1d), consistent with a riboswitch regulation mechanism. To further investigate the mechanism, we engineered a transcriptional construct in which an additional SD sequence was located immediately upstream of *lacZ*, thereby making *lacZ* translation efficiency independent of the riboswitch structure (Fig. 1c). The ß-galactosidase activity was repressed by ~1.6-fold in the presence of TPP in the context of the transcriptional fusion (Fig. 1d). Together, the reporter gene data suggest that TPP sensing by the *tbpA* riboswitch results in the modulation of both levels of translation and transcription.

We next prepared additional constructs to confirm that the regulation is obtained through riboswitch sensing. Given the low expression of the translational and transcriptional fusions, the new constructs contained an arabinose-inducible promoter (pBAD), as previously used to study the *lysC*, *thiM* and *thiC* riboswitches[13,14,22]. As expected, when using the pBAD-driven WT constructs, the ß-galactosidase activity was substantially increased (~10-fold) for both translational and transcriptional fusions (Fig. 1e, f). When these constructs were assessed in the presence of TPP, the expression of *lacZ* was decreased by ~3.2-fold and ~3.9-fold for the translational and transcriptional constructs (Fig. 1e, f), respectively. To ensure that metabolite binding was required for genetic regulation, we introduced a single-point mutation in the TPP binding site (G35C) that is expected to prevent ligand binding[23–25]. Similar levels of *lacZ* expression were obtained with and without TPP for both G35C translational and transcriptional constructs (Fig. 1e, f), consistent with the inability of the riboswitch mutant to regulate gene expression.

Given that the expression of translational fusions is affected by both translational and transcriptional levels, it is difficult to conclude whether the regulatory effects obtained with the translational fusions (Fig. 1e) are primarily due to translational, transcriptional or combined effects. To investigate to which extent the *tbpA* riboswitch rely on translational control to modulate gene expression, we next prepared two different mutants. While the first mutation corresponds to a G121C substitution disrupting the SD sequence, the second one (U130A) transforms the GUG start codon for a GAG sequence, making it translationally inactive. In the context of translational fusions, the SD mutant (G121C) yielded a low *lacZ* expression that was not modulated by the presence of TPP (Fig. 1e). The expression level of the GUG mutant (U130A) was barely detectable, as expected from the inability to initiate translation. In the case of transcriptional fusions, a similarly low level of expression was obtained for the SD mutant when compared to the translational construct (Fig. 1f). However, in contrast to the translational fusion, the GUG mutant resulted in a detectable expression that was repressed by a factor of ~2.4-fold in the presence of TPP (Fig. 1f). These results suggest that *tbpA* mRNA levels can be modulated even in the absence of efficient translational control, similarly to the *lysC* riboswitch[22].

## TPP binding inhibits translation initiation

Since TPP binding to the *tbpA* riboswitch is expected to modulate SD accessibility, we used toeprint assays to determine whether TPP binding inhibits the recruitment of the 30S ribosomal subunit (Fig. 2a). In these assays, nascent *tbpA* transcripts were produced using the *E. coli* RNAP and the binding of the 30S subunit was assessed using primer extension analyses. In the presence of tRNA-fMet and 30S subunit, primer extension assays of the *tbpA* RNA yielded a reversed cDNA product at position C144 (Fig. 2b and Supplementary Fig. 1). This toeprint position agrees well with previous data showing that a P-site tRNA in *E. coli* ribosomes produces a toeprint at +15/16 (where the +1 position is the first nucleotide of the P-site codon)[26].

**Fig. 3 | TPP sensing is preferentially achieved cotranscriptionally by the *tbpA* riboswitch.**
**a** RNase H probing experiments monitoring TPP binding when performed cotranscriptionally (Co-trx) or post-transcriptionally (Post-trx). Control reactions (Ctrl) done in the presence of RNase H (RH) show a cleaved product (P) and the uncleaved full-length (FL) transcript without TPP. Cotranscriptional binding was determined by adding TPP before transcription initiation and performing RNase H cleavage during transcription. Post-transcriptional measurements were achieved by performing transcription without TPP, which was followed by the addition of rifampicin, TPP and RNase H cleavage assays. **b** Quantifications of cotranscriptional and post-transcriptional binding assays. Experiments were fitted to a single-exponential assays and the values of the calculated rates are indicated. **c** $K_{switch}$ measurements for the wild-type *tbpA* riboswitch done in the presence of increasing TPP concentrations. The full-length (FL) and the cleave product (P) are shown on the left of the gel. **d** Quantification of the $K_{switch}$ experiments done when using 50 μM or 500 μM NTP. The $K_{switch}$ value is increasing upon decreasing the NTP concentrations. The average and the standard deviations are shown for each data point.

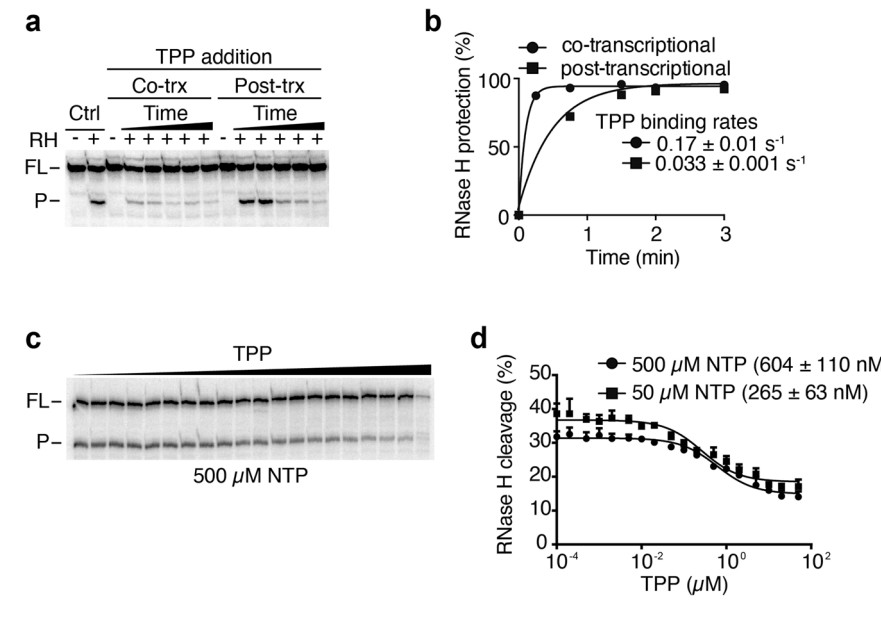

However, the toeprint efficiency was found to decrease by ~2.2-fold in the presence of TPP (Fig. 2b), consistent with TPP binding inhibiting 30S the recruitment of the 30S subunit. No efficient toeprint was observed in the absence of the 30S subunit or tRNA-fMet (Fig. 2b), in agreement with the toeprint requiring a functional initiation complex.

### The *tbpA* riboswitch senses TPP cotranscriptionally

To investigate in more detail the regulation of translation initiation, we next used in vitro transcription-translation assays[14]. In these assays, the first step consists in initiating transcription reactions by adding the *E. coli* RNAP (Fig. 2c). In the second step, transcription reactions are stopped by adding rifampicin and translation reactions are initiated by adding amino acids (Fig. 2c). As a result, these assays effectively permit to uncouple transcription and translation and therefore to determine at which level(s) TPP sensing is achieved. We first assessed the riboswitch regulatory activity by adding TPP during the transcription step, thus allowing cotranscriptional TPP sensing. In such conditions, the expression of ThiC was repressed by ~2-fold (Fig. 2d). We next repeated the experiments by adding TPP post-transcriptionally, i.e., during the translation step. In this case, although ThiC expression was increased by ~1.5-fold, suggesting that the presence of TPP may improve translation efficiency, no repression was detected (Fig. 2d). Control experiments showed that when rifampicin was added at the beginning of transcription reactions, no ThiC product was observed (Fig. 2d), consistent with rifampicin efficiently blocking transcription activity. Furthermore, no TPP-dependent modulation was detected when using *lacZ* (Fig. 2e), consistent with the expression being controlled by the riboswitch. Together, these results indicate that the *tbpA* riboswitch directly modulates the efficiency of translation initiation via the cotranscriptional sensing of TPP.

To investigate in more detail the cotranscriptional metabolite sensing by the riboswitch, we studied the kinetics of TPP binding to the *tbpA* riboswitch using RNase H probing assays[13,14,16,27]. In these experiments, TPP was added either during transcription (cotranscriptionally) or after the addition of heparin (post-transcriptionally)[14]. Control experiments showed that the probe binding to the aptamer can lead to RNase H cleavage in the absence of TPP (Fig. 3a, Ctrl), indicating that this region is accessible in the ligand-free conformation. However, when TPP was added cotranscriptionally, protection from RNase H cleavage was observed at early time points (15 s and 30 s) (Fig. 3a, Co-trX), suggesting efficient TPP binding. In contrast, when TPP was added post-transcriptionally, stronger RNase H cleavages were detected at all time points (Fig. 3a, Post-trX), in agreement with TPP sensing being less efficient. A fitting analysis revealed a fast-binding rate of $0.17 \pm 0.01$ s$^{-1}$ and a slower-binding rate of $0.033 \pm 0.001$ s$^{-1}$ for cotranscriptional and post-transcriptional binding assays, respectively (Fig. 3b). These data suggest that TPP cotranscriptional sensing is performed ~5.7-fold more efficiently than post-transcriptionally, similarly to what reported for the *thiC* riboswitch[14].

### The rate of transcription is important for TPP sensing

Given that our data suggest that the *tbpA* riboswitch preferentially binds to TPP cotranscriptionally to regulate translation (Figs. 2d and 3a), we next examined the importance of the rate of transcription for metabolite sensing[14,16,20,21]. Importantly, due to the presence of three different pause sites located at positions 104, 117 and 136[16], it is hypothesized that such pause sites are important for TPP sensing by allowing more time for cotranscriptional sensing. To study the cotranscriptional binding, we transcribed the *tbpA* riboswitch using various NTP concentrations (500 μM and 50 μM). We reasoned that transcribing at a slower speed would allow more time for TPP binding, thereby decreasing the required TPP concentrations to induce a riboswitch conformational change. When performing RNase H assays as a function of TPP concentration, we found that the cleavage efficiency of RNase H targeting the SD sequence progressively decreased when transcribing with 500 μM NTP (Fig. 3c). Fitting analysis of the experimental data yielded a half TPP concentration ($K_{switch}$) value of $604 \pm 110$ nM for the structural change (Fig. 3d). When the experiments were performed by using a lower NTP concentration (50 μM), the calculated $K_{switch}$ value was $265 \pm 63$ nM (Fig. 3d). These results indicate that a slower rate of transcription leads to a ~2.3-fold reduction in the required TPP concentration to trigger the riboswitch conformational change, consistent with TPP sensing being performed cotranscriptionally. These data are in agreement with our previous study of the *tbpA* riboswitch where the rate of RNAP elongation prior to reaching the pause site 136 modulates TPP sensing[16].

**Fig. 4 | Single-molecule FRET studies of semi-synthetic *tbpA* aptamers. a** Schematic representing the procedure to obtain the semi-synthetic *tbpA* aptamer. A 5' synthetic RNA strand contains Cy3 and Cy5 dyes at the 5' extremity and position 14, respectively. The complete aptamer is reconstituted by ligating the 5' strand and a 3' T7 RNAP-transcribed strand. The aptamer is attached to a PEG slide using a sequence that is hybridized to a DNA anchor containing a biotin. **b** Ligation reactions performed in the presence of the 5' and 3' RNA strands. While the 3' strand corresponds to the 67 nt product, the ligated product of both RNA molecules corresponds to the 107 nt product. **c** smFRET histograms of the semi-synthetic *tbpA* aptamer obtained in the absence and presence of 1 mM TPP. The folded (F) and unfolded (U) states are indicated. Histograms obtained without and with TPP were built using 264 molecules and 323 molecules, respectively. **d** smFRET time trajectories obtained in the absence (*Left*) and presence (*Right*) of TPP. The anti-correlated Cy3 donor and Cy5 acceptor emission intensities are shown with the resulting FRET trace. Photobleaching events are shown by an asterisk. **e** smFRET contour plots obtained in the absence (*Left*) and presence (*Right*) of TPP. Selected traces have been accumulated for the first 50 s. Histograms representing the percentual occupancy of each state are shown to the right. Contour plots obtained without and with TPP were built using 48 molecules and 51 molecules, respectively.

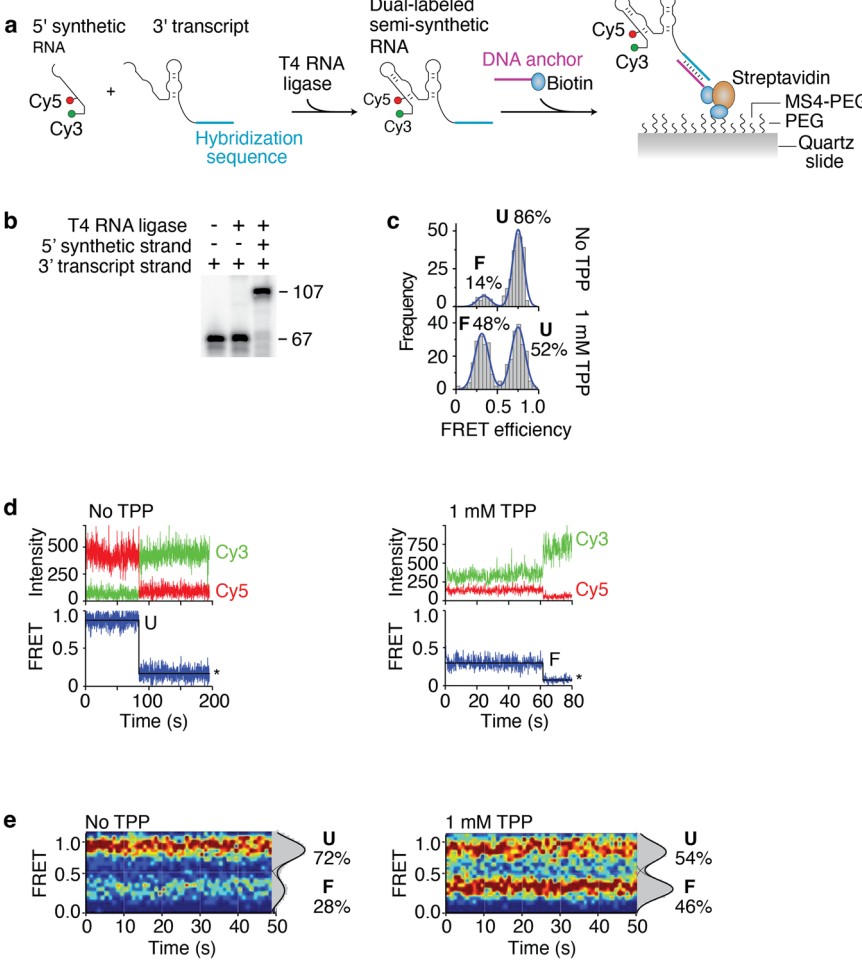

## smFRET analysis of the *tbpA* aptamer

To determine how the structural dynamics of the riboswitch may be used to control the accessibility to the SD region, we employed single-molecule Förster resonance energy transfer (smFRET) to monitor the TPP-induced conformational changes of the aptamer domain[28–30]. Fluorescent donor (Cy3) and acceptor (Cy5) dyes were introduced in the *tbpA* aptamer to follow the RNA conformational changes upon TPP sensing[16]. While the Cy3 dye is located at the 5' extremity of the P1 stem, the Cy5 dye is located at position U14 that is at the interface of the P2 and P3 stems. Because the P3-L5 tertiary interaction is formed in the presence of TPP[23–25], we reasoned that the TPP-dependent structural change should be detected by the smFRET vector defined by positions 1 and 14. To obtain the Cy3-Cy5 dual-labeled *tbpA* aptamer, we relied on a strategy in which a 40-nt 5' synthetic strand containing both Cy3 and Cy5 was ligated to a 67-nt 3' transcript strand (Fig. 4a)[18]. Control experiments showed that both 5' and 3' RNA molecules were efficiently ligated by the T4 RNA ligase (Fig. 4b), thus making the complete *tbpA* aptamer.

To perform smFRET assays, we used a biotinylated DNA anchor that hybridized to the 3' region of the Cy3-Cy5 dual-labeled aptamer allowing to immobilize the molecules on a biotin-functionalized polyethylene glycol surface (Fig. 4a). Single-molecule imaging was performed using wide-field total internal reflection fluorescence microscopy (TIRFM)[31]. In the absence of TPP, the smFRET analysis of the reconstituted *tbpA* aptamer revealed a high fraction of molecules (86%) adopting a high-FRET conformation ($E_{FRET} \sim 0.75$) (Fig. 4c), which corresponds to the previously characterized unfolded (U) state[16]. The aptamer was also found to fold into a minor population (14%) exhibiting a low-FRET conformation ($E_{FRET} \sim 0.3$) that is similar to the TPP-bound folded structure (F state) (Fig. 4c)[16]. In the

presence of 1 mM TPP, the proportion of the F state significantly increased (48%), consistent with the formation of a TPP-*tbpA* aptamer complex (Fig. 4c). The analysis of time traces using hidden Markov modeling without TPP showed long-lived U states (Fig. 4d). Furthermore, a minority of traces revealed that molecules adopted the F state over long period of times or fluctuated between the U and F states (Supplementary Fig. 2a). When the experiments were repeated with TPP, long-lived F states were observed (Fig. 4d) that showed fluctuations between both U and F states (Supplementary Fig. 2b). These results suggest that the U and F conformers show relatively low dynamics during the analyzed timeframe. The distribution of U and F states was obtained by accumulating the FRET time traces from molecules across a time window of ~50 s. While the contour plot obtained in the absence of TPP shows both U (72%) and F (28%) states (Fig. 4e), the contour plot calculated with TPP reveals a significant increase of the F state (46%) across time (Fig. 4e), reflecting the dynamic equilibrium observed in individual traces (Supplementary Fig. 2).

The presence of a substantial fraction (52%) of the unfolded state obtained with the semi-synthetic aptamer in the presence of TPP (Fig. 4c) prompted us to study *tbpA* structural dynamics using nascent transcripts. Due to the inability of the *E. coli* RNAP to incorporate fluorescent nucleotides such as Cy3 or Cy5[16,32], we recently developed an approach to prepare dual-labeled Cy3-Cy5 nascent transcripts obtained with the *E. coli* RNAP[16]. The introduction of Cy3 and Cy5 dyes in transcripts is allowed by initiating transcription with a Cy3-labeled trinucleotide and the site-specific incorporation of an azido-uridine analog. The latter permits the subsequent coupling to a Cy5-alkyne through a click reaction, namely strain-promoted azide-alkyne cycloaddition (SPAAC)[33]. Similarly to the semi-synthetic construct, the aptamer contained a sequence at the 3' extremity that allowed

**Fig. 5 | Single-molecule FRET studies of nascent *tbpA* aptamers obtained by stepwise transcription.** **a** Schematic representing the nascent RNA construct. The nascent RNA has been produced using stepwise transcription reactions and contains a 3' sequence allowing to hybridize to a DNA anchor coupled to a biotin. **b** smFRET histograms of the nascent *tbpA* aptamer obtained in the absence and presence of 1 mM TPP. The folded (F) and unfolded (U) states are indicated. Histograms obtained without and with TPP were built using 210 molecules and 398 molecules, respectively. **c** smFRET time trajectories obtained in the absence (*Left*) and presence (*Right*) of TPP. The anti-correlated Cy3 donor and Cy5 acceptor emission intensities are shown with the resulting FRET trace. Photobleaching events are shown by an asterisk. **d** smFRET contour plots obtained in the absence (*Left*) and presence (*Right*) of TPP. Selected traces have been accumulated for the first 50 s. Histograms representing the percentual occupancy of each state are shown to the right. Contour plots obtained without and with TPP were built using 108 molecules and 117 molecules, respectively.

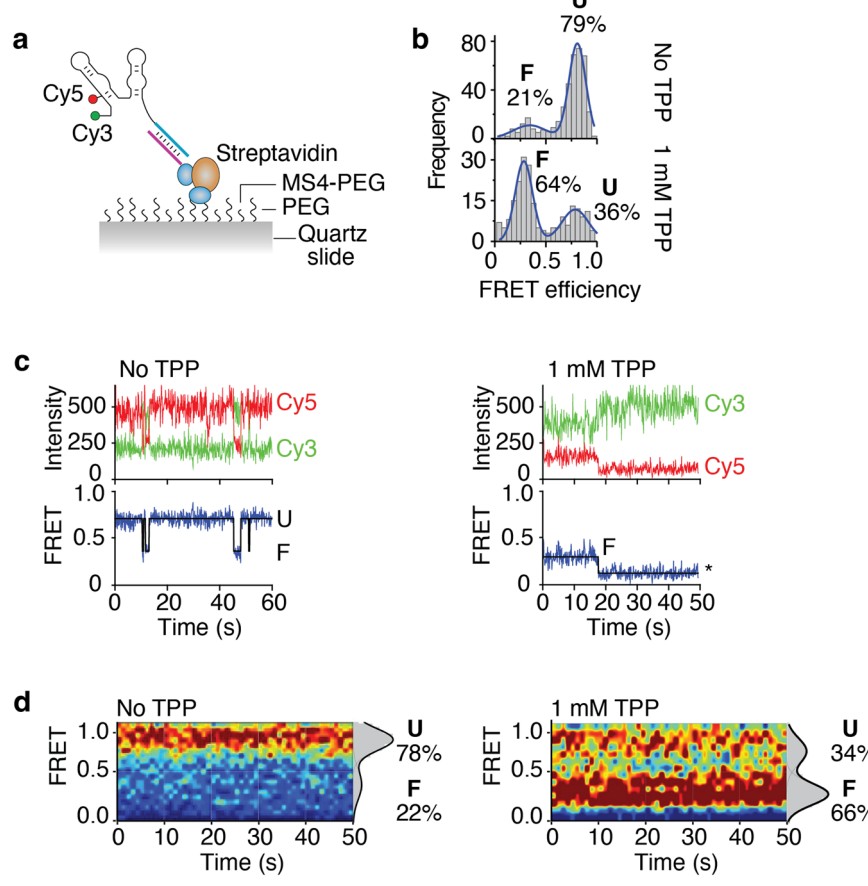

the hybridization of a DNA anchor to immobilize transcripts on a quartz slide (Fig. 5a). Single-molecule imaging showed that Cy3-Cy5 dual-labeled nascent aptamers adopted in a high proportion the U state (79%) in the absence of TPP (Fig. 5b). However, when the experiments were done with 1 mM TPP, the aptamers mostly folded into the F state (64%) (Fig. 5b), which is in contrast to the semi-synthetic construct (Fig. 4c). These results suggest that the use of nascent transcripts allows to obtain better metabolite-dependent folding efficiency. When analyzing individual time traces recorded in the absence of TPP, long-lived U states were observed (Fig. 5c) that were also found to transit to the F state (Supplementary Fig. 3a). In contrast, when doing the experiments with 1 mM TPP, nascent transcripts were found to transit more efficiently to the F state (Fig. 5c) and to exhibit higher dynamics between the U and F states (Supplementary Fig. 3b). The contour plots obtained for a ~50 s timeframe showed that while nascent transcripts remain relatively static in the U state without TPP, they exhibit a larger degree of the F state (66%) in the presence of TPP (Fig. 5d). Together, these results indicate that nascent *tbpA* aptamers show a higher conversion of the U to F state in the presence of TPP when compared to the renatured construct.

## Discussion

Our study suggests that the *E. coli* TPP-sensing *tbpA* riboswitch regulates gene expression both by modulating translation initiation and mRNA levels, similarly to other *E. coli* riboswitches sensing TPP (*thiC* and *thiM*)[13,14], lysine (*lysC*)[22,34] and FMN (*ribB*)[35]. Importantly, the secondary structure of the *tbpA* riboswitch is very similar to that of the *thiM* riboswitch, suggesting that a related regulation mechanism is expected. However, in contrast to the *thiM* riboswitch in which 34 codons are necessary to downregulate mRNA levels upon TPP sensing[13], our data showed that only 12 codons are sufficient to obtain significant regulation (Fig. 1f). These results indicate that a smaller portion of *tbpA* 5' UTR is required to modulate mRNA levels

compared to *thiM*. Thus, both *tbpA* and *thiM* may rely on part of the coding region to modulate mRNA levels, which is in contrast to *thiC* and *lysC* that only require the aptamer domain[14,22]. Both *thiC* and *lysC* riboswitches are larger than *tbpA* and *thiM*, suggesting that they embed the structural information to control mRNA levels using Rho or RNase E[14,22]. More work will be required to understand molecular details of *tbpA* translational and transcriptional regulatory mechanisms. For instance, it was recently proposed that TPP sensing by the *tbpA* riboswitch represses gene expression at the translational level, but not at the transcriptional level[17]. While these experiments ruled out a Rho-dependent transcription termination mechanism using bicyclomycin, it is still possible that mRNA levels could be modulated by other factors, such as RNase E. Furthermore, given that a shorter construct (4 codons) was used in that study, it is conceivable that regulatory elements contained within the downstream 8 codons sequence are missing. If this is the case, the regulation at the RNA level would be triggered because of translation inhibition. Interestingly, our data suggest that the TPP-dependent RNA regulation may be obtained in the absence of active translation elongation (Fig. 1f, TrX construct, GUG mutant). However, since no mRNA regulation was observed when mutating the SD sequence (Fig. 1f, TrX constructs, SD mutant), it indicates that the binding of the ribosome at the SD sequence is required for the mRNA regulation to take place. Thus, although these results indicate that no active translation is needed to control the mRNA levels, they suggest that ribosome binding is still required for the mRNA regulation.

Our toeprint analysis using *tbpA* nascent transcripts revealed that TPP binding could sterically modulate the access of the 30S ribosomal subunit (Fig. 2b), consistent with the riboswitch regulation mechanism. The in vitro uncoupling of transcription and translation further showed that the riboswitch translation regulation relied on the cotranscriptional TPP sensing (Fig. 2d), similar to the *thiC* riboswitch[14]. The preference for ligand sensing during transcription elongation was also observed when monitoring

kinetics of RNase H cleavage reactions (Fig. 3a). Using a probe binding to the aptamer domain, the TPP binding rate was found to be ~5.7-fold more efficient when achieved under cotranscriptional conditions than post-transcriptionally (Fig. 3b), a condition similar to the *thiC* riboswitch[14]. Furthermore, transcription reactions performed at different elongation rates showed that the $K_{switch}$ value decreased by ~2.3-fold when transcription reactions were done lower nucleotide concentrations (Fig. 3d). The preference for TPP binding under cotranscriptional conditions suggest that TPP sensing is performed within a defined transcriptional window, i.e., when the RNAP is transcribing a specific locus of the 5' UTR. While the upstream boundary for TPP binding should correspond to the formation of the aptamer domain, the downstream boundary is less clear. However, it was previously determined using smFRET assays that transcription complexes located at the pause site 136 are refractory to TPP binding[16]. These results suggest that at position 136, the formation of the anti-P1 stem most likely occurs (Fig. 1b), which is expected to disrupt the aptamer domain and therefore to perturb ligand binding. Consequently, the formation of the anti-P1 stem is predicted to define the downstream boundary of the TPP sensing window. This situation is similar to the *thiC* riboswitch for which the formation of the anti-P1 stem is also involved in the formation of a structure precluding TPP sensing[14]. Since it was recently determined that an intermediate structure embedded within the anti-P1 stem is involved in the cotranscriptional folding of the *tbpA* riboswitch[17], it suggests that different mechanisms of cotranscriptional folding are used by *tbpA* and *thiC* riboswitches.

Our single-molecule FRET analysis of the *tbpA* aptamer was done using two versions of the *tbpA* riboswitch: semi-synthetic and nascent RNA constructs. The semi-synthetic construct allows to position the Cy3 and Cy5 dyes at any positions during the chemical RNA synthesis. However, due to its inherent nature, it does not allow to study nascent RNA molecules, which are likely representing more biologically relevant structures. The nascent RNA constructs rely on the use of stepwise transcription and click chemistry reaction to introduce a Cy3 dye at the 5' end of the transcript and a Cy5 dye at an uracil residue. Despite the fundamental differences between these two approaches, both techniques revealed that the *tbpA* aptamer adopted two FRET states, U and F, and that the F state was favored in the presence of TPP (Figs. 4c and 5b). The main difference observed between both constructs lies in the proportion of F state obtained in the presence of TPP, which is higher for the nascent RNA construct. These results indicate that the use of nascent transcripts allows to ensure better folding efficiency upon metabolite binding. Specifically, the higher efficiency of nascent transcripts is consistent with their folding being adopted cotranscriptionally, which has been shown to be crucial for the adoption of native RNA structures[36]. In contrast, due to the nature of their preparation, semi-synthetic RNA aptamers are denatured and renatured prior to smFRET assays, thus possibly allowing for partially misfolded structures and hence lower TPP binding efficiency[36,37].

Furthermore, our smFRET data show that relatively low structural dynamics of the *tbpA* aptamer in the absence of TPP for both constructs, which is mostly stabilized in the U state (Figs. 4e and 5d). In the presence of TPP, while a greater proportion of the F state is observed, a relatively low dynamic is occurring between both U and F state (Figs. 4e and 5d). Such a low structural exchange could be due to the absence of the expression platform, which should allow competition between the ON and OFF states[14,16]. Therefore, these smFRET data are consistent with the *tbpA* aptamer exhibiting a large structural change in the presence of TPP but in which a low level of dynamics is observed between both FRET states. While more work is required to decipher the molecular mechanism between the U and F conformational states, previous data indicate that higher rates of structural exchanges may be obtained in the context of the full *tbpA* riboswitch in which the anti-P1 stem could be formed[16]. Thus, in a cotranscriptional context, it is conceivable that exchange between both states occurs within the transcriptional window in which the anti-P1 stem

may partially be completed, most particularly when intermediate structures are formed[17].

Overall, our findings are consistent with the *tbpA* riboswitch performing a dual modulation of gene expression by controlling both translation initiation and mRNA levels, which is similar to other *E. coli* riboswitches[13,14,22,34,35]. Such a control of transcriptional levels allows to regulate the expression of several genes embedded within downstream operons, thus permitting to efficiently adjust the production of involved metabolic pathways using additional regulatory mechanisms such as Rho-dependent transcription termination[13,35]. For example, when sensing TPP, the *E. coli tbpA*, *thiC* and *thiM* riboswitches[38] collectively downregulate the mRNA levels of 11 genes involved either in the transport or biosynthesis of TPP. While such an expanded regulatory effect is expected from transcriptionally-regulating riboswitches, such as in *B. subtilis*, our findings add new data for the *tbpA* riboswitch supporting the idea that translationally-regulating *E. coli* riboswitches also modulate whole operon mRNAs upon metabolite sensing. The ability for riboswitches to regulate mRNA levels is possible due to their propensity to sense metabolites cotranscriptionally[14,16,17,35,39]. The higher efficiency of cotranscriptional sensing for translational riboswitches implies that metabolite recognition is achieved by nascent transcripts, which is supported by our RNase H and smFRET data (Figs. 3d and 5b). Lastly, given that the selective sequestration of the Shine-Dalgarno sequence may be performed post-transcriptionally, such riboswitches may also control outside of the cotranscriptional binding window and benefit from an additional layer of genetic control allowing for fine tuning regulation[20]. Clearly, the cotranscriptional sensing of cellular metabolites is at the heart of both transcriptional and translational riboswitches and therefore suggests that the transcriptional process is essential for riboswitch regulation.

In conclusion, our work showed that the *E. coli tbpA* riboswitch may regulate both at the translational and transcriptional levels. Furthermore, the requirement for cotranscriptional TPP sensing emphasizes the presence of a mRNA regulation mechanism that is also observed in other *E. coli* riboswitches. Our single-molecule data is consistent with nascent transcripts being more efficient to sense TPP, a situation that is consistent with riboswitches performing efficient genetic response during transcription elongation.

## Methods
### Bacterial strains and oligonucleotides
Strains used in this study are derived from *Escherichia coli* MG1655. Strain BL21 (DE3) was used for overproduction of RNA polymerase (RNAP) and sigma70 factor. The *E. coli* RNAP and sigma70 factors were obtained from the *Plateforme de purification des protéines* (Université de Sherbrooke). The PM1205 strain (Supplementary Table 1) was used to construct the *tbpA* translational and natural promoter fusions (Supplementary Table 2)[22]. The PCR strategies to obtain the various constructs are listed in Supplementary Table 3. DNA oligonucleotides used in this study were purchased from Integrated DNA technologies (IDT) and are listed in Supplementary Table 4.

### Beta-galactosidase experiments
Kinetic assays for beta-galactosidase experiments were performed as described previously[14,22]. Briefly, an overnight bacterial culture grown in M63 0.2% glucose minimal medium was diluted to an $OD_{600}$ of 0.02 in 50 mL of fresh medium. The various fusions used in this study are described in the Supplementary Table 2. For constructs containing the natural promoter, the culture was incubated at 37°C until an $OD_{600}$ of 0.1 was obtained, and TPP (500 µg/mL) was then added as indicated. For constructs using a pBAD promoter, arabinose (0.1% final concentration) was added after $OD_{600}$ of 0.1 was reached, which was followed by the addition of TPP (500 µg/mL). In all cases, the specific ß-galactosidase activities were calculated as previously described[22,40].

### In vitro transcription-translation assay

The transcription-translation assays were performed using the template *pLacUV5-tbpA₃₀₆cd* (Supplementary Table 3) and were done as previously reported[14]. Briefly, reactions were achieved using the PURExpress Kit from New England Biolabs. To verify that TPP does not generally affect transcription-translation efficiency, a DNA template containing the lacUV5 promoter fused to the first 934 codons of *lacZ* followed by a T7 terminator was used as an internal control (*pLacUV5-lacZ*). TPP was added at the final concentration of 25 µM and transcription was carried out with 0.5 U of the RNA polymerase holoenzyme from Epicentre. For *tbpA* uncoupled transcription-translation assays, reactions were initiated by a transcription step of the *pLacUV5-tbpA₃₀₆cd* construct at 37 °C for 15 min using a transcription solution without amino acids/transfer RNA. Rifampicin (250 µg/mL) was next added to the reaction and incubated for 1 min to prevent re-initiation of transcription. The translation step was initiated by incubating with a solution comprising amino acids and transfer RNA at 37 °C for 4 h. TPP (100 µM) was either added during the transcription step (cotranscriptional) or the translation step (post-transcriptional). Reactions were stopped by placing samples on ice for 10 min and next incubated with 4 volumes of acetone 100% at 4 °C for 15 min. Samples were precipitated, resuspended in denaturing buffer, and resolved on SDS-PAGE.

### RNase H probing analysis of transcription reactions

DNA templates corresponding to *tbpA* EC-248 (*pLacUV5-tbpA-EC-248*) were incubated with sigma70 factor, the RNAP, and the streptavidin at 37 °C for 5 min. Transcription reactions were performed in 20 mM Tris-HCl pH 8.0, 20 mM MgCl₂, 20 mM NaCl, 14 mM 2-mercaptoethanol and 0.1 mM EDTA. Elongation complexes were synchronized at position 9 (EC-9) by incorporating 10 µM GUU trinucleotide, 2.5 µM ATP/CTP, [α-³²P] UTP and incubating at 37 °C for 5 min. Sample were passed through G50 columns to remove free nucleotides. Samples were added to a reaction mixture composed of TPP and all four nucleotides, and were incubated at 37 °C for 5 min to elongate at position 248. Transcriptions were completed at various TPP concentrations (1 nM to 500 µM) and NTP concentrations (50 µM and 500 µM). After incubating at 37 °C for 5 min, samples were mixed with 200 µM DNA probe (923AC) and incubated for another 5 min. RNase H cleavage assays were initiated by incorporating a solution of RNase H in 5 mM Tris-HCl, pH 8.0, 20 mM MgCl₂, 100 mM KCl and 5 µM EDTA and incubated at 37 °C for 5 min. Reactions were stopped by adding an equal volume of stop solution (95% formamide, 20 mM EDTA and 0.4% SDS).

### RNase H probing analysis of TPP kinetics

The kinetics of TPP binding (co- or post-transcriptional binding) was essentially performed as previously described[14]. Transcription reactions using the *tbpA* EC-136 DNA template (*pLacUV5-tbpA-EC-136*) were performed in 20 mM Tris-HCl pH 8.0, 20 mM MgCl₂, 20 mM NaCl, 14 mM 2-mercaptoethanol and 0.1 mM EDTA. Radioactively labeled transcripts were prepared as described in the section "RNase H probing analysis of transcription reactions". TPP (10 µM) was either added cotranscriptionally or post-transcriptionally by adding an excess of heparin prior to TPP[14]. The use of an excess of heparin was shown to prevent RNAP from initiating transcription[41], thus allowing to monitor post-transcriptional TPP binding on transcribed RNA molecules. The obtained transcripts were assessed by RNase H cleavage assays by mixing with 200 µM DNA (24477AC) and RNase H for 15 s at various time points (15 s, 45 s, 90 s, 2 min and 3 min) (see section "RNase H probing analysis of transcription reactions" for RNase cleavage assays). The reported errors for the TPP-binding rates are the standard error in the fitting[18], which are assumed to be approximated by the standard deviation.

### TPP-dependent toeprint assays

Nascent 180 nt *tbpA* transcripts were prepared by in vitro transcription reactions using the EC-180 DNA template (*pLacUV5-tbpA-EC-180*) in 20 mM Tris-HCl pH 8.0, 20 mM MgCl₂, 20 mM NaCl, 14 mM 2-mercaptoethanol and 0.1 mM EDTA. Reactions were initiated by

incubating the DNA template, sigma70 factor and the RNAP at 37 °C for 5 min. Elongation complexes were synchronized at position 9 (EC-9) by incorporating 25 µM Cy3-GUU trinucleotide, 2.5 µM ATP/CTP/UTP and incubating at 37 °C for 5 min. Sample were passed through G50 columns to remove free nucleotides. Where indicated, 500 µM TPP was added followed by the addition of 500 µM NTPs. Reactions were incubated at 37 °C for 10 min and the obtained RNAs were purified using the ThermoFisher GeneJET RNA purification kit. The obtained transcripts were incubated with the radiolabeled oligonucleotide (3374JG) in 50 mM Tris-HCl pH 8.3, 75 mM KCl, 3 mM MgCl₂ and 10 mM DTT at 37 °C for 5 min. Next, 300 nM of the 30S subunit, 600 nM tRNA-fMet (MP Biomedicals) and 1 mM TPP and incubated at 37 °C for 5 min. Then, 500 µM dNTPs and 10 U of MulV-RT (New England Biolabs) were added and the reverse transcriptase reaction allowed to proceed at 37 °C for 15 min. Reactions were migrated on an 8% denaturing-PAGE for 2 h and products were quantified by using the Quantity One software. The intensity of the toeprint was normalized to a band that did not show any change across conditions (C138).

Sequencing reactions used for the analysis of reverse transcription products were done using RNA produced using T7 RNAP transcription reactions[42] using the *pT7-tbpA-EC-180* template. Briefly, 100 nM of RNA was incubated with 250 µM ddNTPs in TE 1X buffer (10 mM Tris-HCl and 1 mM EDTA). The radiolabeled oligonucleotide (3374JG) was incubated with the RNA at 65 °C for 3 min and cooled down at room temperature for 5 min. A mix of SS first strand buffer (Invitrogen), 5 mM DTT and 500 µM dNTPs were added to the mixture following the addition of the Superscript II RT enzyme (Invitrogen) and incubated at 52 °C for 10 min.

### Preparation of semi-synthetic *tbpA* RNA for smFRET assays

The Cy3-Cy5 dual-labeled *tbpA* aptamer was prepared using ligation assays[18]. Briefly, the aptamer was reconstituted using a synthetic 5' strand (IDT) (residues 1–38) and a transcribed 3' strand (residues 39 to 89). The 5' RNA strand was ordered as a 5'-labeled Cy3 strand containing an azide-modified U at position 14 (2558JG). The 5' RNA strand was incubated at 37 °C for 1 h with 63 µM µM DBCO-Cy5 (Jean Bioscience) in 20 mM Tris-HCl pH 8.0, 20 mM MgCl₂, 20 mM NaCl, 14 mM 2-mercaptoethanol and 0.1 mM EDTA to obtain a Cy3-Cy5 labeled strand, which was purified by denaturing gel electrophoresis and ethanol precipitation. The 3' strand was produced using T7 RNAP transcription reactions[42] using the 3' *tbpA-89* template. The 3' strand was obtained through self-cleavage of a larger transcript containing a hammerhead ribozyme located upstream of the aptamer RNA sequence. Transcription reactions were resolved using denaturing polyacrylamide gel electrophoresis and the band corresponding to the 3' strand was excised from the gel and purified using ethanol precipitation[18]. The obtained 3' strand was subsequently phosphorylated using T4 Polynucleotide Kinase (NEB) using the manufacturer's protocol. Next, the Cy3-Cy5 dual-labeled 5' strand and the 5'-PO₄ 3' transcribed strand were annealed by heating a mixture (molar ratio 1:1) to 80 °C in 50 mM Tris-HCl pH 7.5 and 50 mM NaCl and slowly cooling it to room temperature. The mixture was transferred in a tube containing 50 mM Tris-HCl pH 7.5, 10 mM MgCl₂, 1 mM DTT and 1 mM ATP and T4 RNA ligase 1 (New England Biolabs), which was incubated at 37 °C for 4 h. Full-length ligated RNA molecules were purified by denaturing polyacrylamide gel electrophoresis and ethanol precipitation.

### Preparation of nascent *tbpA* RNA for smFRET assays

The Cy3-Cy5 dual-labeled nascent *tbpA* aptamers were prepared using native transcription reactions[16]. Briefly, DNA templates allowing the transcription of the *tbpA* aptamer + anchoring sequence (*pLacUV5-tbpA-88-extBio*) were incubated with sigma70 factor and the RNAP at 37 °C for 5 min. Transcription reactions were done in 20 mM Tris-HCl pH 8.0, 20 mM MgCl₂, 20 mM NaCl, 14 mM 2-mercaptoethanol and 0.1 mM EDTA. Elongation complexes were synchronized at position 9 (EC-9) by incorporating 25 µM Cy3-GUU trinucleotide, 1.25 µM ATP/CTP/UTP and incubating at 37 °C for 5 min. Sample was passed through G50 columns to

remove free nucleotides. Samples were reacted with 2.5 μM CTP/GTP and 2.5 μM azide-modified UTP analog at 37 °C for 5 min to allow formation of EC-17. The samples were next passed through G50 columns to remove free nucleotides. The full-length transcripts were obtained by adding 1 mm NTP at 37 °C for 5 min, thus releasing the RNA from transcription complexes. The incorporation of Cy5 at the U14 azide group was performed by incubating transcripts with 63 μM DBCO-Cy5 in the transcription buffer at 37 °C for 1 h. The transcripts were passed through G50 columns to remove unreacted DBCO-Cy5 dyes. Samples were collected in the transcription buffer and were incubated 5 min with or without TPP before smFRET measurements.

### Single-molecule FRET imaging

Single-molecule FRET experiments were performed as previously described[16,18]. smFRET traces were recorded from immobilized single *tbpA* aptamers using a prism-type total-internal reflection setup including an inverted microscope (Olympus IX71) coupled to a 532 nm laser (Crystalaser) and a back illuminated Ixon EMCCD camera (Andor). Microscope quartz slides were passivated with a 30:1 mixture of PEG (Laysan Biosciences, USA) and biotinylated PEG. A concentration of 0.2 mg/mL streptavidin was added to the PEG slide to allow immobilization of the *tbpA* aptamer hybridized with the biotinylated DNA anchor. A concentration of ~250 pM dual-labeled nascent or semi-synthetic *tbpA* aptamers was added to the slide and attached to the slide using DNA anchors 2433AC or 2559JG, respectively. Data were acquired using a 50 ms integration time on surface-immobilized molecules in the transcription buffer to which was added 2 mM Trolox, 5 mM 3,4-protocatechuic acid (PCA, Sigma) and 100 nM protocatechuate dioxygenase (PCD, Sigma), pH 7.5, which was used to enhance the photostability of the dye. The setup allowed to record Cy3 and Cy5 signals simultaneously[16,18]. Left and right images were corrected for optical aberrations and inhomogeneous evanescence-wave illumination using custom-built routines using IDL software (Exelis, USA). Subsequently, an IDL mapping algorithm was used to correlate the position of Cy3 and Cy5 signals from the same molecule and a time trace was built for each molecule. To calculate the percentual contribution of the U and F states at each experimental condition, the single-molecule FRET histogram at each condition was fitted to a two-Gaussian distribution. The area under each Gaussian was extracted and its percentual contribution calculated with respect to the total area under the two-Gaussian curve.

### Statistics and reproducibility

All the experiments were repeated for two or three times. For reporter gene assays, independent batches of culture were included in each experiment.

### Reporting summary

Further information on research design is available in the Nature Portfolio Reporting Summary linked to this article.

### Data availability

The data that support the findings of this study are available from the corresponding author upon reasonable request. Source data underlying Figs. 1–3 can be found in the Supplementary Data. The non-cropped version of gels are presented in Supplementary Fig. 4 and 5.

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

## Acknowledgements

We thank members of the Lafontaine and Penedo laboratory for discussion. This work was supported by the Canadian Institutes of Health Research, the Natural Sciences and Engineering Research Council of Canada (NSERC) and the Fonds de recherche du Québec en Natures et Technologies (FRQNT). J.C.P. thanks the School of Biology at the University of St Andrews for financial support.

## Author contributions

Conceived, designed, and performed experiments: J.P.G, M.G., M.S.-R., A.C., P.T., P.S.-P., A.D., J.M. Analyzed the data: J.P.G, M.G., M.S.-R., A.C., P.T., P.S.-P., A.D., J.M., E.M., J.C.P. and D.A.L. Participated in writing the manuscript: J.P.G, P.S.-P., J.C.P. and D.A.L.

## Competing interests

The authors declare no competing interests.
