## [Peer Review file · Communications Biology]

Insights into the Cotranscriptional and Translational Control Mechanisms of the *Escherichia coli* *tbpA* thiamin pyrophosphate riboswitch.

Corresponding Author: Professor Daniel Lafontaine

Version 0:

Reviewer comments:

Reviewer #1

(Remarks to the Author)

Grondin et al demonstrate that the *E. coli* thiamin pyrophosphate (TPP) *tbpA* riboswitch controls protein expression at both transcriptional and translational levels. They also present several lines of evidence that thiamin pyrophosphate sensing is more effective when it occurs co-transcriptionally. They use single-molecule methods to show that aptamer domain spontaneously samples folded and unfolded conformations in the absence of the ligand. The authors use an impressive array of experimental approaches. Most of their conclusions are convincing. The manuscript is suitable for publication pending few revisions and modifications that are needed to address several experimental issues. Conceptually, the authors could elaborate more on the broad implications of their studies for the mechanisms of riboswitch control of gene expression.

Experimental issues:

- 1) Method section is rather abbreviated and lacks important details. For example, buffer conditions for ribosome-mRNA complex assembly in toeprinting analysis are not specified. The source of tRNA^{fMet} is not mentioned. Buffer conditions should be spelled out for every experiment described in the paper.
- 2) Ribosome-independent band in toeprinting analysis that runs right above of the ribosome toeprint (C144) gets darker in the presence of initiator tRNA. This raises a doubt whether C144 band is the authentic ribosome toeprint. The size of toeprint seems right though as P-site tRNA in *E. coli* ribosome typically produces the toeprint at +15/16 position (where +1 is the first nucleotide of the P-site codon, e.g. Joseph&Noller, EMBO J 1998). The authors might want to state that in the text of the manuscript in support their interpretation of toeprint data. Still, quality of toeprinting data shown in Suppl Fig 1 is questionable: there are many ribosome-independent bands, which might be products of RNase-induced degradation of mRNA. The authors might be able to improve quality of toeprinting data by repeating these experiments in the presence of RNase inhibitors and/or using more homogeneous mRNA preparations.
- 3) There are numerous issues in single-molecule tracking of DNA oligo binding to the riboswitch. Heat maps in Fig. 4 e-f show only marginal difference in binding of Cy3-labeled oligo with fluorescence intensities above 200 a.u. threshold in the absence and presence of TPP. Fig. 4h show similarly broad distributions of dwell times in experiments done in the absence and presence of TPP. 200 au threshold cut-off seems arbitrary. Slightly longer dwell times for both bound and unbound states observed in the absence of TPP might simply be due to having significantly more binding events in "no TPP" dataset. Equilibrium binding constants (the ratio of 1/"bound" and 1/"unbound" dwell times) are similar in the absence and presence of TTP that contradicts authors' conclusions (Suppl. Fig 3). Single-molecule measurements of fluorescence intensities for a single fluorophore are always complicated by spurious signals and intensity fluctuations (by contrast, smFRET measurements based on donor and acceptor intensities are much more reliable because of anti-correlated changes in two fluorophores). Many of the transient binding events might be non-specific. Overall, I do not find these single-molecule data and their analysis reliable and compelling. I would advocate for removing Fig. 4 from the manuscript. In my opinion, main conclusions of the manuscript will be unaffected by this change.
- 4) smFRET data of Fig. 5 and 6 are quite compelling. However, it is not quite clear why FRET distribution histograms for the experiments done in the presence TPP shown in Fig. 6b and 6d appreciably differ in the fraction of U (high FRET) state. This is important because the authors use comparisons of Fig. 5c and 6b to argue that TTP sensing is more effective when it occurs co-transcriptionally. However, contrary to this interpretation, histograms obtained in the presence of TPP, which are shown in Fig. 5e and Fig. 6d, are quite similar to each other. The authors should explain this discrepancy and/or tone down their conclusions about co-transcriptional TPP sensing at least in the context of smFRET data.

Reviewer #2

(Remarks to the Author)

The manuscript authored by Grondin and colleagues presents interesting and comprehensive co-transcriptional and post-transcriptional analyses of TPP riboswitch in *E. coli*. The authors discovered that the riboswitch exerts its regulatory influence on both transcription and translation, with greater efficacy co-transcriptionally compared to post-transcriptionally. The experimental methodology employed appears to be meticulous, accompanied by the necessary controls. The introduction is concise, and the discussion facilitates comprehension.

Post-transcriptional data were procured by adding heparin to inhibit transcription. However, the experimental evidence to elucidate the role of heparin in the TPP riboswitch's transcription is missing.

In Fig. 3b, only two or three data points were utilized to calculate the binding rates, which is insufficient. A more comprehensive dataset should be acquired before equilibrium of the RNaseH-cleavage.

The rationale for employing a cut-off intensity value of 200 counts to define the bound and unbound states (Line 236) is missing.

Regarding the smFRET data, the authors claimed a nascent riboswitch was used in Fig. 6, whereas a post-transcriptional riboswitch was used in Fig. 5. Nevertheless, as no FRET data was collected in the presence of *E. coli* RNAP, it suggests that the RNA used in both figures represents post-transcriptional RNA. Clarification is needed regarding the significant differences observed between Fig. 5 and Fig. 6.

Author Rebuttal letter:

Notes:

1. All corrections performed in the manuscript are highlighted using the "Track changes" function in Word. The number of some figures has been changed from the original manuscript version.

Referee 1

Grondin et al demonstrate that the *E. coli* thiamin pyrophosphate (TPP) *tbpA* riboswitch controls protein expression at both transcriptional and translational levels. They also present several lines of evidence that thiamin pyrophosphate sensing is more effective when it occurs co-transcriptionally. They use single-molecule methods to show that aptamer domain spontaneously samples folded and unfolded conformations in the absence of the ligand. The authors use an impressive array of experimental approaches. Most of their conclusions are convincing. The manuscript is suitable for publication pending few revisions and modifications that are needed to address several experimental issues. Conceptually, the authors could elaborate more on the broad implications of their studies for the mechanisms of riboswitch control of gene expression.

Ref1, Concern 1.

Method section is rather abbreviated and lacks important details. For example, buffer conditions for ribosome-mRNA complex assembly in toeprinting analysis are not specified. The source of tRNA^{fMet} is not mentioned. Buffer conditions should be spelled out for every experiment described in the paper.

Response by Authors: We thank the Referee for this comment. We have updated the Methods section by adding more experimental details such as the buffer compositions. We also have expanded the description about the preparation of fluorescent aptamers.

Regarding the comment about the broad implications of our findings, we have rewritten the Discussion to take this into account. For instance, we have removed the first paragraph and used some of the ideas of that paragraph to create a new one that was added at a later stage of the Discussion. We have added this text (p. 18, line 354): "Overall, our findings are consistent with the *tbpA* riboswitch performing a dual modulation of gene expression by controlling both translation initiation and mRNA levels, which is similar to other *E. coli* riboswitches^{13,14,22,34,35}. Such a control of transcriptional levels allows to regulate the expression of several genes embedded within downstream operons, thus permitting to efficiently adjust the production of involved metabolic pathways using additional regulatory mechanisms such as Rho-dependent transcription termination^{13,35}. For example, when sensing TPP, the *E. coli* *tbpA*, *thiC* and *thiM* riboswitches³⁸ collectively downregulate the mRNA levels of 11 genes involved either in the transport or biosynthesis of TPP. While such an expanded regulatory effect is expected from transcriptionally-regulating riboswitches, such as in *B. subtilis*, our findings add new data for the *tbpA* riboswitch supporting the idea that translationally-regulating *E. coli* riboswitches also modulate whole operon mRNAs upon metabolite sensing. The ability for riboswitches to regulate mRNA levels is possible due to their propensity to sense metabolites cotranscriptionally^{14,16,17,35,39}. The higher efficiency of cotranscriptional sensing for translational riboswitches implies that metabolite recognition is achieved by nascent transcripts, which is supported by our RNase H and smFRET data (Fig. 3d, 5b). Lastly, given that the selective sequestration of the Shine-Dalgarno sequence may be performed post-transcriptionally, such riboswitches may also control outside of

the cotranscriptional binding window and benefit from an additional layer of genetic control

Grondin et al., 2024—Response to referees 1/5

allowing for fine tuning regulation²⁰. Clearly, the cotranscriptional sensing of cellular metabolites is at the heart of both transcriptional and translational riboswitches and therefore suggests that the transcriptional process is essential for riboswitch regulation."

Ref1, Concern 2.

Ribosome-independent band in toeprinting analysis that runs right above of the ribosome toeprint (C144) gets darker in the presence of initiator tRNA. This raises a doubt whether C144 band is the authentic ribosome toeprint. The size of toeprint seems right though as P-site tRNA in *E. coli* ribosome typically produces the toeprint at +15/16 position (where +1 is the first nucleotide of the P-site codon, e.g. Joseph & Noller, EMBO J 1998). The authors might want to state that in the text of the manuscript in support their interpretation of toeprint data. Still, quality of toeprinting data shown in Suppl Fig 1 is questionable: there are many ribosome-independent bands, which might be products of RNase-induced degradation of mRNA. The authors might be able to improve quality of toeprinting data by repeating these experiments in the presence of RNase inhibitors and/or using more homogeneous mRNA preparations.

Response by Authors: To further support our data, we have rephrased the part describing the toeprint and added the relevant reference, as suggested by the Referee, at page 9 line 153: "In the presence of tRNA-fMet and 30S subunit, primer extension assays of the *tbpA* RNA yielded a reversed cDNA product at position C144 (Fig. 2b and Supplementary Fig. 1). This toeprint position agrees well with previous data showing that a P-site tRNA in *E. coli* ribosomes produces a toeprint at +15/16 (where the +1 position is the first nucleotide of the P-site codon²⁶)."

Regarding the additional ribosome-independent bands, we agree with the Referee that it could suggest the presence of impurities or RNases in the reaction. However, in contrast to toeprint assays performed using T7 RNAP transcripts, our toeprints analysis was done using nascent transcripts obtained with the *E. coli* RNAP. In our laboratory, we typically observe that reverse transcription reactions performed using such nascent *E. coli* RNAP transcripts generate intermediate cDNA species at a higher frequency (Chauvier et al., Nat Commun 2017). Although we never formally investigated this aspect, we hypothesize that nascent *E. coli* RNAP transcripts may exhibit more complex structures than renatured transcripts obtained with T7 RNAP. Importantly, the toeprint at position C144 is observed even in the presence of such additional ribosome-independent reverse transcriptase products, which agrees with the *tbpA* riboswitch modulating translation initiation.

Ref1, Concern 3.

There are numerous issues in single-molecule tracking of DNA oligo binding to the riboswitch. Heat maps in Fig. 4 e-f show only marginal difference in binding of Cy3-labeled oligo with fluorescence intensities above 200 a.u. threshold in the absence and presence of TPP. Fig. 4h show similarly broad distributions of dwell times in experiments done in the absence and presence of TPP. 200 au threshold cut-off seems arbitrary. Slightly longer dwell times for both bound and unbound states observed in the absence of TPP might simply be due to having significantly more binding events in "no TPP" dataset. Equilibrium binding constants (the ratio of 1/"bound" and 1/"unbound" dwell times) are similar in the absence and presence of TPP that contradicts authors' conclusions (Suppl. Fig 3). Single-molecule measurements of fluorescence intensities for a single

Grondin et al., 2024—Response to referees 2/5

fluorophore are always complicated by spurious signals and intensity fluctuations (by contrast, smFRET measurements based on donor and acceptor intensities are much more reliable because of anti-correlated changes in two fluorophores). Many of the transient binding events might be non-specific. Overall, I do not find these single-molecule data and their analysis reliable and compelling. I would advocate for removing Fig. 4 from the manuscript. In my opinion, main conclusions of the manuscript will be unaffected by this change.

Response by Authors: As suggested by the Referee, we have removed Figure 4 (and related data in Supplementary Figures 2 and 3) from the paper. As a result, we have rewritten some parts of the paper that were referring to these experiments. In addition, we have chosen a new title that better suits the resulting paper.

Ref1, Concern 4.

smFRET data of Fig. 5 and 6 are quite compelling. However, it is not quite clear why FRET distribution histograms for the experiments done in the presence TPP shown in Fig. 6b and 6d appreciably differ in the fraction of U (high FRET) state. This is important because the authors use comparisons of Fig. 5c and 6b to argue that TPP sensing is more effective when it occurs co-transcriptionally. However, contrary to this interpretation, histograms obtained in the presence of TPP, which are shown in Fig. 5e and Fig. 6d, are quite similar to each other. The authors should explain this discrepancy and/or tone down their conclusions about co-transcriptional TPP sensing at least in the context of smFRET data.

Response by Authors: We are grateful to the referee for spotting this discrepancy between the relative smFRET populations of the U and F states and we agree this is crucial to support our conclusions. We have carefully revised our raw data and realized that the smFRET histogram shown in Figure 6d (Figure 5d in the new manuscript version) with TPP did not correspond to the state 1 mM concentration but to a lower concentration. We apologize for this mistake and we have now replaced the contour plot and corresponding smFRET histogram with the correct 1 mM TPP concentration (Figure 5d, right panel). In addition to provide a more quantitative comparison of the relative populations of U and F for each construct and conditions, we have now added percentual contributions for each FRET population. These values were calculated by quantifying the area under the Gaussian curve corresponding to the U and F populations taking as 100% of the total area (U + F). These percentual contributions have now been added in Figures 4 and 5 next to the corresponding FRET state and mentioned across the manuscript where relevant. In support of our conclusions, the nascent ttpA aptamer shows a change in the relative contributions of the F state from 21%/22% (Figure 5b and Figure 5d, no TPP) to a value of ~64%/66% in presence of 1 mM TPP (Figures 5b and 5d, 1 mM TPP). We have included the following sentence in the Materials and Methods to explain the calculation of the percentual contributions of each state (p. 26, line 539): "To calculate the percentual contribution of the U and F states at each experimental condition, the single-molecule FRET histogram at each condition was fitted to a two-Gaussian distribution. The area under each Gaussian was extracted and its percentual contribution calculated with respect to the total area under the two-Gaussian curve."

Grondin et al., 2024—Response to referees 3/5

Referee 2

The manuscript authored by Grondin and colleagues presents interesting and comprehensive co-transcriptional and post-transcriptional analyses of TPP riboswitch in *E. coli*. The authors discovered that the riboswitch exerts its regulatory influence on both transcription and translation, with greater efficacy co-transcriptionally compared to post-transcriptionally. The experimental methodology employed appears to be meticulous, accompanied by the necessary controls. The introduction is concise, and the discussion facilitates comprehension.

Ref2, Concern 1.

Post-transcriptional data were procured by adding heparin to inhibit transcription. However, the experimental evidence to elucidate the role of heparin in the TPP riboswitch's transcription is missing.

Response by Authors: We thank the Referee for pointing out the incomplete description regarding the use of the polyanion heparin. The latter has been shown to inhibit free RNAP to perform transcription elongation (Coupar and Chesterton, *Eur. J Biochem* 1977; Chamberlin et al., *JBC* 1979). Because of the polyanionic character of heparin, it mimics DNA molecules and can thus bind to free RNAP that are released upon finishing the transcription of DNA templates, therefore blocking transcription. Similarly to the assays shown in this manuscript, we have previously used heparin to monitor TPP binding both co- and post-transcriptionally using RNase H assays for the *E. coli* thiC riboswitch (Chauvier et al., *Nat Commun* 2017). In these assays, an excess of heparin was added prior to TPP to block transcription, which thus allowed to monitor post-transcriptional TPP binding on transcribed RNA molecules. We have clarified the use of heparin in the Methods section "RNase H probing analysis of TPP kinetics" (p. 21, line 422): "TPP (10 μ M) was either added cotranscriptionally or post-transcriptionally by adding an excess of heparin prior to TPP14. The use of an excess of heparin was shown to prevent RNAP from initiating transcription³⁹, thus allowing to monitor post-transcriptional TPP binding on transcribed RNA molecules."

Ref2, Concern 2.

In Fig. 3b, only two or three data points were utilized to calculate the binding rates, which is insufficient. A more comprehensive dataset should be acquired before equilibrium of the RNase H-cleavage.

Response by Authors: The results shown in Figure 3b were obtained by monitoring the TPP-bound state of radioactive transcripts using RNase H assays at different time points. In these

experiments, the RNase cleavage reaction time of 5 min has been decreased to 15 s to allow shorter time points to be analyzed. This is important since transcription reactions are still active in the co-transcriptional conditions and longer RNase H reaction time would preclude to monitor short time points (below 1 min). Importantly, it is difficult to interpret shorter time (<30 s) points due to the RNase H reaction time. Nevertheless, our claim is supported by the time required to achieve 90% RNase H protection between cotranscriptional (30 s) and post-transcriptional (~1.5 min), which corresponds to a ~5-fold variation.

It would be very difficult for our laboratory to investigate this aspect since we no longer have the facilities to radioactive-based experiments. While we agree with the Referee that additional points

Grondin et al., 2024—Response to referees 4/5
would strengthen our conclusions, we still think that the obtained data is relevant in suggesting that co-transcriptional sensing is more efficient than when done post-transcriptionally.

Ref2, Concern 3.
The rationale for employing a cut-off intensity value of 200 counts to define the bound and unbound states (Line 236) is missing.

Response by Authors: Due to various points raised by the Referee 1, we have removed the data presented in Fig. 4. As a result, we have rewritten some parts of the paper that were referring to these experiments. In addition, we have chosen a new title that better suits the resulting paper.

Ref2, Concern 4.
Regarding the smFRET data, the authors claimed a nascent riboswitch was used in Fig. 6, whereas a post-transcriptional riboswitch was used in Fig. 5. Nevertheless, as no FRET data was collected in the presence of *E. coli* RNAP, it suggests that the RNA used in both figures represents post-transcriptional RNA. Clarification is needed regarding the significant differences observed between Fig. 5 and Fig. 6.

Response by Authors: In this study, we have prepared Cy3-Cy5 fluorescent *tbpA* aptamers using two approaches. Fluorescent aptamers were obtained either through transcription reactions using *E. coli* RNAP or through the ligation of synthetic RNA strands. As the Referee is correctly indicating, both smFRET assays were performed in the absence of RNAP and aptamer folding was monitored post-transcriptionally. However, in the presence of the *E. coli* RNAP, the sensing of TPP by the riboswitch is expected to be done during the transcription elongation process, which is essentially performed cotranscriptionally.

The point of comparison that is made in the manuscript is that nascent RNA aptamers obtained using the *E. coli* RNAP exhibit a different smFRET folding pattern compared to their semi-synthetic version. The higher proportion of the F state with TPP exhibited by nascent transcripts suggests that aptamers are more efficiently binding TPP. We reasoned that the efficient TPP binding activity of nascent aptamers is most probably due to cotranscriptional folding. In contrast, due to the nature of their preparation, semi-synthetic RNA aptamers were denatured and renatured prior to smFRET assays, thus possibly allowing for partially misfolded structures and hence a lower TPP binding efficiency.

We have expressed our conclusions more clearly about this in the Discussion (p. 17, line 334):
“These results indicate that the use of nascent transcripts allows to ensure better folding efficiency upon metabolite binding. Specifically, the higher efficiency of nascent transcripts is consistent with their folding being adopted cotranscriptionally, which has been shown to be crucial for the adoption of native RNA structures³⁶. In contrast, due to the nature of their preparation, semi-synthetic RNA aptamers are denatured and renatured prior to smFRET assays, thus possibly allowing for partially misfolded structures and hence lower TPP binding efficiency.”

Grondin et al., 2024—Response to referees 5/5

Version 1:

Reviewer comments:

Reviewer #1

(Remarks to the Author)

The authors have comprehensively addressed all issues raised in previous reviews. I believe that this manuscript should be accepted for publication in its current form.

Reviewer #2

(Remarks to the Author)

The manuscript has been substantially revised and is now suitable for publication in Communications Biology.
